# Antibacterial Activity and Mechanism of Madecassic Acid against *Staphylococcus aureus*

**DOI:** 10.3390/molecules28041895

**Published:** 2023-02-16

**Authors:** Chunling Wei, Peiwu Cui, Xiangqian Liu

**Affiliations:** 1College of Pharmacy, Hunan University of Chinese Medicine, Changsha 410208, China; 2Research Lab of TCM Property & Efficacy, Level 3, National Administration of TCM, Changsha 410208, China; 3Mycomedicine Research Lab, Hunan University of Chinese Medicine, Changsha 410208, China

**Keywords:** madecassic acid, *Staphylococcus aureus*, antibacterial activity, antibacterial mechanism

## Abstract

Antibacterial resistance has become one of the most serious problems threating global health. To overcome this urgent problem, many scientists have paid great attention to developing new antibacterial drugs from natural products. Hence, for exploring new antibacterial drugs from Chinese medicine, a series of experiments were carried out for verifying and elucidating the antibacterial activity and mechanisms of madecassic acid (MA), which is an active triterpenoid compound isolated from the traditional Chinese medicine, *Centella asiatica*. The antibacterial activity was investigated through measuring the diameter of the inhibition zone, the minimum inhibitory concentration (MIC), the growth curve, and the effect on the bacterial biofilm, respectively. Meanwhile, the antibacterial mechanism was also discussed from the aspects of cell wall integrity variation, cell membrane permeability, and the activities of related enzymes in the respiratory metabolic pathway before and after the intervention by MA. The results showed that MA had an inhibitory effect on eight kinds of pathogenic bacteria, and the MIC values for *Staphylococcus aureus*, Methicillin-resistant *Staphylococcus aureus*, *Escherichia coli*, *Pseudomonas aeruginosa*, *Bacillus subtilis*, and *Bacillus megaterium* were 31.25, 62.5, 250, 125, 62.5, and 62.5 µg/mL, respectively. For instance, 31.25 µg/mL MA could inhibit the growth of *Staphylococcus aureus* within 28 h. The antibacterial mechanism experiments confirmed that MA could destroy the integrity of the cell membrane and cell wall of *Staphylococcus aureus*, causing the leakage of macromolecular substances, inhibiting the synthesis of soluble proteins, reducing the activities of succinate dehydrogenase and malate dehydrogenase, and interacting with DNA, leading to the relaxation and ring opening of supercoiled DNA. Besides, the activities of DNA topoisomerase I and II were both inhibited by MA, which led to the cell growth of *Staphylococcus aureus* being repressed. This study provides a theoretical basis and reference for the application of MA in the control and inhibition of food-borne *Staphylococcus aureus*.

## 1. Introduction

As one of the most common causes responsible for infectious diseases and an important pathogen in clinical practice, *Staphylococcus aureus* (*S. aureus*) has contributed to a severe threat, involving mild skin infection, severe tissue infection, and sepsis to human health due to its widely distribution in human skin, especially in the nasopharynx [1]. In addition, *S. aureus* can also cause a variety of nosocomial infections derived from medical instruments and equipment pollution [2]. At present, the treatment of *S. aureus* infection mainly relies on antibiotics. However, repeated and excessive use of antibiotics has led to serious antibiotic residues and bacterial resistance, along with decreased therapeutic efficacy. Therefore, it has become an important and urgent task to explore natural products with low toxicity and high anti-microbial activity for treating the initial stage of infection, reducing the use of antibiotics, and limiting the development of drug resistance of pathogenic microorganisms. Besides, active natural compounds can also perform as adjuvants or bacterial resistance-modifying agents, which can enhance or restore the effectiveness of commercial drugs towards antibiotic-resistance bacteria, so it has drawn much awareness to mine potential natural compounds with significant antibacterial activity from pharmaceutical botany.

*Centella asiatica* (L.) Urb is a famous botanical drug, belonging to the umbrella family, recorded in the 2020 edition of the Chinese Pharmacopoeia [3], which is widely distributed in China, India, Malaysia, Indonesia, Oceania Islands, Japan, Australia, and Central Africa [4,5]. In China, the main producing area of *Centella asiatica* (L.) Urb is located in Anhui, Guangxi, Hunan, and other central south places. The dried whole of this plant can be drunk as herbal tea [6] due to its effects of clearing heat and dampness, detoxification and swelling [7], and showing cold and bitter taste. Relevant studies have also shown that *Centella asiatica* can exhibit antibacterial properties, liver injury protection, nerve protection, anti-tumor properties, wound healing properties, and other effects [8,9,10,11,12], indicating that *Centella asiatica* (L.) Urb is a good source for developing new antibacterial drugs. As one of the most active constituents in *Centella asiatica* (L.) Urb, asiatic acid can show significant broad-spectrum antibacterial activity on *Escherichia coli*, *Salmonella typhimurium*, *Pseudomonas aeruginosa*, *Enterococcus faecalis*, *S. aureus*, etc. [13,14,15,16]. However, our research confirmed that another main compound in *Centella asiatica* (L.) Urb, MA, also showed significant antibacterial activity. As MA has been reported to exhibit anti-cancer properties, cardiovascular protection, anti-diabetes properties, and anti-inflammatory activities in previous studies [17,18,19,20], it can be recognized as a competitive and promising candidate compound for drug development. Based on this, the antimicrobial activity and related mechanism of MA were systematically studied in this paper, which can provide an integrated experimental basis and reference for the development of low-toxicity and low-resistance antimicrobial drugs.

## 2. Results

### 2.1. Antibacterial Activity of MA

#### 2.1.1. Diameter of Inhibition Zone

The inhibition zone of oxacillin (OXA) against *S. aureus* was the largest, and no antibacterial zone was formed in the dimethyl sulfoxide (DMSO, control) group. The inhibition effect of MA on *S. aureus* was better, and the diameters of inhibition zones of MA against *S. aureus*, methicillin-resistant *Staphylococcus aureus* (MRSA), *Candida albicans* (*C. albicans*), *Escherichia coli* (*E. coli*), *Pseudomonas aeruginosa* (*P. aeruginosa*), *Gordinia* sp., *Bacillus subtilis* (*B. subtilis*), and *Bacillus megaterium* (*B. magaterium*) were 13 mm, 14.5 mm, 14 mm, 10 mm, 11.5 mm, 13.5 mm, 10.5 mm, and 10.5 mm, respectively (Table 1), indicating that MA is a potential candidate compound for anti-pathogenic bacteria drug development.

#### 2.1.2. Minimum Inhibitory Concentration (MIC)

The MIC values of MA against *S. aureus*, MRSA, *E. coli*, *P. aeruginosa*, *B. subtilis*, and *B. magaterium* were 31.25, 62.5, 250 and 125, 62.5, and 62.5 µg/mL, respectively (Table 1). The MIC of OXA (positive control) against *S. aureus*, MRSA, *B. subtilis*, and *B. magaterium* were 0.048, 7.8, 3.9, and 1.9 µg/mL, respectively. The experimental results showed that the antibacterial effect of MA against Gram-positive bacteria (*S. aureus*, MRSA) was better than that of Gram-negative bacteria (*E. coli*, *P. aeruginosa*). It was also found that the MIC of MA against MRSA was stronger than that of tormentic acid (128 μg/mL) [21], but lower than that of sanguisorbigenin (12.5–50 μg/mL) [22]. Even MA, tormentic acid and sanguisorbigenin all belong to ursane-type triterpenoids, and they exhibit different inhibition activity against MRSA, which may be caused by the different substituent group of different compounds.

### 2.2. Antibacterial Effect of MA on S. aureus

In comparison to the control group (DMSO), the treatment with 31.25 µg/mL and 62.5 µg/mL MA significantly inhibits the growth of S. aureus. The optical density (OD) value of the MA group was significantly lower than that of the control group. This further confirmed that MA had an obvious inhibitory effect on *S. aureus*, and 31.25 µg/mL MA could exhibit consistent growth inhibiting activity of *S. aureus*, up to 28 h, while 62.5 µg/mL MA could completely inhibit the growth of *S. aureus* within 32 h (Figure 1).

### 2.3. Effect of MA on Biofilm of S. aureus

A bacterial biofilm is an organized bacterial group formed by multiple bacteria adhering to non-biological or biological surfaces, secreting polymer matrix, and wrapping themselves in it [23]. The results of crystal violet staining showed that, when the concentration of MA was 31.25 µg/mL and 62.5 µg/mL, the inhibition rates of biofilm were 68.35% and 72.73%, respectively, indicating higher concentration of MA, causing inhibiting rates of increasing biofilm (Figure 2). Therefore, MA had an inhibitory effect on the production of *S. aureus* biofilms.

### 2.4. Effect of MA on Cell Membrane and Cell Wall of S. aureus

#### 2.4.1. Results of Changes in Conductivity in the Culture Medium after the Intervention of MA on *S. aureus*


The experimental group with MA showed a rapid growth trend of broth conductivity, and the percentage of electrical conductivity for the MIC group and the 2 MIC group reached 67.17% and 76.79%, respectively, while the control group was only 36.32% (Figure 3a). This result suggests that MA has a significant impact on the function of the cell membrane of *S. aureus*, which is the main barrier preventing the leakage of cell contents.

#### 2.4.2. Effect of MA on Macromolecular Substances in Culture Medium of *S. aureus*


After 31.25 µg/mL and 62.5 µg/mL MA treatment, the nucleic acid leakage (OD_260nm_) and protein leakage (OD_280nm_) in the extracellular bacteria suspension were significantly increased (Figure 3b,c). The results also indicated that the permeability of the cell membrane increased after the action of MA, and the macromolecular substances in the cell were released into the extracellular broth.

#### 2.4.3. Effect of MA on Cell Wall of *S. aureus*

The leakage amount of alkaline phosphatase in the experimental group containing MA was significantly higher than that in the control group (Figure 4), indicating that MA could increase the leakage amount of alkaline phosphatase (AKP) in *S. aureus*, and the integrity of the bacteria wall might be damaged, symbolizing that MA could damage the cell wall of *S. aureus*, causing the leakage of cell contents.

#### 2.4.4. Effect of MA on β-Galactosidase Content

*β*-Galactosidase exists in the bacterial inner membrane. When the inner membrane is damaged, *β*-Galactosidase would leak out through the cytoplasmic membrane, and the change in extracellular *β*-galactosidase content could be used to predict the damage degree of bacterial inner membrane [24,25]. *β*-galactosidase content of the control group did not change significantly during the whole cultivation period, but *β*-galactosidase content of the MAs, 31.25 µg/mL and 62.5 µg/mL experimental groups, was significantly higher than that of the control group (Figure 5), indicating that MA destroyed the integrity of the inner membrane of *S. aureus* and caused the leakage of *β*-galactosidase to the extracellular surface, leading to an increase in extracellular *β*-galactosidase.

### 2.5. Effect of MA on Soluble Protein of S. aureus

The results shown in Figure 6, which were analyzed by sodium dodecyl sulfate polyacrylamide gel electrophoresis (SDS-PAGE) of soluble protein separated from the cells of *S. aureus*, the control group without MA treatment presented more and clearer protein bands than the two experimental group intervened by 31.25 µg/mL MA and 62.5 µg/mL MA, respectively. This result suggests that MA could exert its antibacterial effect through repressing the protein expression in *S. aureus*. The reason may be that MA affects the synthesis of nucleic acid and the expression of related genes of *S. aureus*, and then it blocks the synthesis of some proteins and enzymes, leading to a decrease in protein expression in the bacteria, thus exerting its bacteriostatic effect.

### 2.6. Study on the Interaction Mode between MA and DNA

#### 2.6.1. Changes of DNA Structures by Interaction of MA with DNA

Plasmid DNA was chosen to observe the MA effects on the superhelical structure of circled double-stranded DNA. The results of agarose gel electrophoresis are shown in Figure 7. When the concentration of MA was greater than 0.007812 mg/mL, Form I (supercoiled DNA) gradually decreased, and Form II (relaxation unwinding DNA) gradually increased, indicating that MA could cause relaxation and rang opening of super spiral DNA.

#### 2.6.2. UV Spectral Study on the Binding of MA with DNA

Ultraviolet absorption spectroscopy is often used to study the interaction between compounds and DNA. Generally, red shift phenomenon and color reduction effect appeared after the interaction between compounds and DNA, indicating that the interaction between the compounds and DNA is intercalation binding [26]. With the increase in MA concentration, the maximum light absorption value of pET-28a DNA gradually decreased, and the color reduction effect occurred. The maximum adsorption wavelength of pET-28a DNA showed a red shift, which further proved that MA could bind pET-28A DNA, and DNA conformational changes were caused by the separation of DNA base pairs caused by MA insertion (Figure 8).

### 2.7. Effect of MA on the Morphology of S. aureus Cells

The results of transmission electron microscopy showed that MA had an obvious effect on the morphology of *S. aureus*. The untreated *S. aureus* had normal morphology and an intact membrane. However, compared with the untreated *S. aureus*, the cell plasma membrane of the *S. aureus* exposed to 31.25 µg/mL MA was damaged, and the surface was rougher. The cells of *S. aureus* exposed to 62.5 µg/mL MA were spited, and cytoplasmic contents were released (Figure 9).

### 2.8. Effect of MA on the Determination of Malate Dehydrogenase (MDH) and Succinate Dehydrogenase (SDH) Activities

MDH and SDH are the key enzymes in the tricarboxylic acid cycle. Detecting the activities of MDH and SDH can indirectly reflect the energy metabolism in bacteria. The activities of SDH and MDH in *S. aureus* were significantly decreased after exposure to 31.25 µg/mL and 62.5 µg/mL MA (Figure 10). This suggests that MA could inhibit the activities of MDH and SDH in *S. aureus*.

### 2.9. Effect of MA on DNA Topoisomerase Activity of S. aureus

#### 2.9.1. Purity Test Results of pET-28a

The agarose gel electrophoresis pattern is shown in Figure 11. pET-28a is a commonly used prokaryotic efficient expression vector of fusion protein type, the size of which is 5369 bp, containing kanamycin resistance gene. The electrophoresis bands showed that the size of purified plasmid DNA was between 5000 bp and 6000 bp coupled with a brighter band referred to the supercoiled pattern of target plasmid.

#### 2.9.2. Determination of DNA Topoisomerase Activity in Crude Enzyme Extracts of *S. aureus*


The supercoiled pET-28a DNA can be uncoiled into open loop or linear DNA (Form II) by the crude enzyme isolated from the cells of *S. aureus*. With the increase in concentration of crude enzyme, Form II DNA gradually increased, along with a gradual decrease in supercoiled DNA (Form I), indicating that the DNA topoisomerase in the crude enzyme is effective and has strong unwinding activity (Figure 12).

#### 2.9.3. Effects of MA on DNA Topoisomerase I Activity of *S. aureus*

DNA topoisomerase I is one of the key enzymes in nucleic acid metabolism in organism catalyzing transient DNA single-strand disconnection and reconnection related to DNA unwinding regulation occurring in the process of DNA replication, transcription, recombination, reparation, and other key reactions. This process does not require the participation of energy cofactor adenosine triphosphate (ATP). Under the catalysis of DNA topoisomerase I, pET-28a DNA was unscrewed into Form II. When the concentration of MA was greater than 1.25 mg/mL, the Form I gradually increased, and Form II gradually decreased with the increase in MA concentration, which indicated that MA could inhibit the activity of *S. aureus* topoisomerase I (Figure 13).

#### 2.9.4. Effects of MA on DNA Topoisomerase II Activity of *S. aureus*

DNA topoisomerase II is another important target enzyme responsible for breaking two strands of a double helix of DNA. The role of topoisomerase II is to mediate the unwinding, breaking, and reconnecting of DNA double strands to affect the structure of DNA. When the concentration of MA was greater than 2.5 mg/mL, the Form I gradually increased, and the Form II gradually decreased with the increase in MA concentration, indicating that MA could inhibit the activity of topoisomerase II from *S. aureus* (Figure 14).

## 3. Discussion

*S. aureus*, a widely distributed pathogenic bacteria, can cause local suppurative infections generating blood infection, pneumonia, enteritis, and other suppurative inflammation related diseases in human. To cure these infections, many and excessive antibiotics and other drugs have been adopted to these patients, but the bacterial resistance has also been induced at the same time, which has become a new deep, challenging issue threating human health. 

In order to slow down or repress the evolution of bacterial resistance, the development of new antibacterial Chinese herbal medicines and their preparations have become an important hotspot in the field of antibacterial drug research and development. For example, Zhang et al. [27] found that the aqueous extract of *Artemisia argyi* leaves could destroy the integrity of the cell wall of *S. aureus* and increase the permeability of the cell membrane at a certain concentration, but it could not kill *S. aureus* in a short time; Zhou et al. [28] found that a part of n-butanol and ethanol crude extract of the *Dendrobium* shell had strong antibacterial activity against *Salmonella paratyphimurium* and *S. aureus*, and the antibacterial mechanism might be related to the synthesis inhabitation of bacterial proteins. However, both Chinese herbal compound preparations and single Chinese herbal medicines all have complex chemical components, leading to unclear antibacterial mechanisms and lack of standard drug use specifications. Therefore, it is necessary to conduct in-depth research on the antibacterial activities and mechanisms of key components in Chinese herbal medicines. Relevant data will provide an important basis for the development of antibacterial drugs from Chinese herbal medicines and the formulation of clinical drug use specifications.

In this study, the filter paper method and the 2,3, 5-triphenyltetrazolium chloride (TTC) staining method were used to determine the antibacterial activity and MIC value of MA. The antibacterial activity of MA against eight common pathogenic bacteria (*S. aureus*, MRSA, *C.albicans*, *E. coli*, *P. aeruginosa*, *Gordinia* sp., *B. subtilis*, and *B. magaterium*) was determined. Results showed that the antibacterial effect of MA against Gram-positive bacteria was better than that of Gram-negative bacteria. The MIC of MA against *S. aureus* was 31.25 μg/mL. The growth curves of 31.25 µg/mL and 62.5 µg/mL MA against *S. aureus* were also determined. It was further confirmed that MA had a significant inhibitory effect on the growth of *S. aureus* and could inhibit the growth of *S. aureus* within 28 h. Meanwhile, 31.25 µg/mL and 62.5 µg/mL MA could significantly inhibit the formation of *S. aureus* biofilm. Bacteria cells are composed of cell membrane, cell wall, cytoplasm, and nucleoplast, among which cell membrane is an important medium for material and energy exchange between bacteria and the outside world. When the cell membrane permeability of pathogenic bacteria is destroyed, the electrolytes in the bacteria will leak into the culture medium, and the conductivity of the supernatant of the culture medium of *S. aureus* can indirectly reflect the permeability of the cell membrane of the bacteria. In this study, the change in the relative conductivity of the cell after the action of MA was detected. It was found that the relative electrical conductivity of the experimental group was significantly higher than that of the control group after 4 h of MA treatment, which indicated that the permeability of the cell membrane of *S. aureus* was changed after MA treatment, causing the leakage of small molecules, such as K^+^ and Na^+^ in the cells. Furthermore, OD values of nucleic acid leakage (OD_260nm_) and protein leakage (OD_280nm_) in extracellular bacterial suspensions were detected. It was found that the OD values of nucleic acid leakage and protein leakage of bacterial suspensions increased significantly after MA treatment. This further confirmed that the cell membrane of *S. aureus* was damaged after the action of MA, and the macromolecular substances, such as DNA and RNA in the cell, were released into the culture medium, thus leading to the death of *S. aureus*. At the same time, the effect of MA on *β*-galactosidase content was also measured, and it was found that the extracellular *β*-galactosidase content of MA treatment group was significantly higher than that of the control group, which fully confirmed that MA damaged the cell membrane of *S. aureus*. AKP is an enzyme that exists between cell walls. Under normal circumstances, the activity of AKP could not be detected outside the cell wall, but when the cell wall was damaged, AKP could be detected outside the cell wall, and the activity of AKP could also increase [29]. The experimental results showed that the content of AKP in the experimental group was significantly higher than that in the control group, indicating that MA could destroy the bacterial cell wall and increase the permeability of the bacterial cell wall. The experimental results showed that the content of AKP in the experimental group was significantly higher than that in the control group, indicating that MA could destroy the bacterial cell wall and increase the permeability of the bacterial cell wall. At the same time, MA could also inhibit the synthesis of soluble protein of *S. aureus*. Besides, the morphology of *S. aureus* cells treated with 31.25 µg/mL and 62.5 µg/mL MA was observed by transmission electron microscope. The result is consistent with the previous data on the effects of MA on the cell wall and the membrane showed above. SDH and MDH are the key metabolic enzymes in the tricarboxylic acid cycle of pathogenic bacteria. Detection of SDH and MDH activities in *S. aureus* can reflect the energy metabolism of *S. aureus*. In this study, the activities of SDH and MDH in *S. aureus* before and after MA intervening were detected, and it was found that MA could reduce the activities of SDH and MDH in *S. aureus*. This means that MA plays an antibacterial role by acting on metabolic enzymes in *S. aureus*. In addition, the method of interaction between MA and DNA was also tested to determine whether drugs could interact with DNA. Agarose gel electrophoresis results showed that the interaction between MA and DNA caused relaxation and looped opening of supercoiled DNA, and UV absorption spectroscopy further proved that MA could bind pET-28a DNA. The antibacterial mechanism of MA is from our research, as shown in Figure 15. As antibiotic resistance has become the global challenge threating people’s health, and many pathogenic bacteria export antibiotics from intracellular environment through developing their multidrug efflux pumps system, which can be induced or evolved under long term excessive drug stress by modifying or regulating relative genes in plasmids [30], it is very urgent to develop new drugs or adjuvants to overcome antibiotic resistance. In this study, the multi-mechanical actions against *S. aureus* by MA were systemically verified, indicating that MA is a promising natural compound for controlling *S. aureus* infection. In further study, how MA regulates or interacts with the multidrug efflux pump system of pathogenic bacteria is also a very important issue. So, this research can also provide a basic reference for further elucidating the molecular mechanism of MA against pathogenic bacteria. 

## 4. Methods

### 4.1. Bacterial Strains and Bacterial Culture

*S. aureus*, CMCC (B) 26003 (Shanghai Luwei Technology Co., Ltd.), MRSA ATCC 43300 (BeiJing Microbiological Culture Collection Center (BJMCC)), and other strains, including *Escherichia coli* (*E. coli*) ATCC 25922, *Pseudomonas aeruginosa* (*P. aeruginosa*) ATCC 27853, *Candida albicans* (*C. albicans*) CMCC (F) 98001, *Bacillus megaterium* (*B. magaterium*) ATCC35985, and *Bacillus subtilis* (*B. subtilis*), were all purchased from Shanghai Bioresource Collection Center, and the strain *Gordinia* sp. JD-4 was isolated from the local soil by our lab and identified by the 16s rDNA sequence. For Madecassic acid (Figure 16), the purity by HPLC analysis was more than 98%. Lot number: J12HB184580, Shanghai Yuanye Biotechnology Co., Ltd.; Luria Bertani (LB) agar medium, LB liquid medium, yeast extract peptone dextrose (YPD) agar medium, and YPD liquid medium, Guangdong Huankai Microbiology Technology Co., Ltd.

### 4.2. Determination of Diameter of Inhibition Zone of MA

The paper diffusion method [31] was employed to evaluate the antibacterial activity of MA. Briefly, the strains were firstly, respectively, activated on the ultraclean table, and they were uniformly coated on the LB medium plate with sterile coating rods. On the plate, 6 mm filter paper sheets soaked with MA solution of certain concentration were placed, respectively; another solvent-only sample was selected as the control group. After inoculation and filter paper plating, the plates were cultured in the biochemical incubator at 37 °C for 12–18 h. Finally, the antibacterial effect was calculated through measuring the diameter of the inhibition zone by Vernier caliper, which is a measuring tool for measuring length, inner and outer diameter, and depth accurately.

### 4.3. Determination of Minimum Inhibitory Concentration (MIC)

The MIC values of MA against six strains (*S. aureus*, MRSA, *E. coli*, *P. aeruginosa*, *B. subtilis* and *B. magnatum*) were determined by the TTC staining method [32]. After dilution of the bacterial suspension in the logarithmic phase to the concentration of 10^6^–10^7^ cfu/mL by liquid medium, 180μL bacterial suspension and 20μL MA solution of different concentrations were added to the 96-well plate to ensure the final concentrations of 500, 250, 125, 62.5, 31.25, 15.625, 7.8, and 3.9 µg/mL, respectively. DMSO was chosen as the control group, and the plate was incubated at 37 °C for 12–16 h. After incubation, TTC solution was added to each experimental well, and the plate was incubated at 37 °C for another 4 h in the dark for observing the color change. The experiment was repeated three times, and the average value was calculated.

### 4.4. Antibacterial Curve of MA against S. aureus

Different volumes of MA solution were added into 10^7^–10^8^ cfu/mL *S. aureus* suspension to reach the final concentrations of 31.25 µg/mL and 62.5 µg/mL. The control group was added DMSO. Then, all the culture tubes were transferred to a constant temperature culture oscillator with parameter settings at 37 °C and 120 rpm. During the cultivation period, OD at 600 nm was measured every 4 h. Finally, the growth curves of *S. aureus* with or without MA treatment was drawn to show the relationship between the culture time (t) and the OD value [33]. The experiment was repeated thrice, and the average value was taken.

### 4.5. Inhibitory Effect of MA on the Production of S. aureus Biofilm

*S. aureus* was inoculated into LB medium and incubated at 37 °C, 120 rpm for about 24 h. When the OD_600nm_ of the bacteria broth reached about 0.3, the MA solution of different concentrations was added to the bacteria broth and mixed thoroughly to reach the final MA concentration of 31.25 µg/mL and 62.5 µg/mL, respectively. In the control group, equal volume of DMSO was added to the broth. After 24 h cultivation at 37 °C, the planktonic cells were removed and washed by phosphate buffered saline (PBS) thrice. Subsequently, the samples were dyed with 1% crystal violet for 10 min at room temperature, then washed for three times by sterile water. Finally, the stained biofilm was dissolved in 200 μL absolute ethanol, and the OD value at 570 nm was measured [34].

### 4.6. Transmission Electron Microscope (TEM) Analysis of the Effect of MA on the Morphology of S. aureus Cells

TEM observation was employed to analyze the effect of MA on the morphology of *S. aureus* cells. Firstly, *S. aureus* cells were cultivated to the exponential phase to prepare the bacterial suspensions. Then, MA was added to reach the final concentration of 31.25 µg/mL and 62.5 µg/mL, and the obtained bacterial suspensions were incubated for another 6 h at 37 °C. The control group was treated with equivalent DMSO. After incubation, the cells were washed with sterilized PBS, processed with 2.5% glutaraldehyde, and fixed with 1% osmium tetroxide for 24 h, and then they were dehydrated with a gradient ethanol concentration. Finally, the cells were embedded in resin for ultrathin sections, and the samples were observed and photographed with a transmission electron microscope for analyzing the changes in intracellular structure of *S. aureus* [35].

### 4.7. Determination of Electrical Conductivity, DNA, RNA, and Other Macromolecular Substances Extravasation and Alkaline Phosphatase in Bacterial Liquid Phase

The electrical conductivity was measured according to the methods reported previously [36]. The suspension of *S. aureus* in the logarithmic growth phase was centrifuged for 10 min and washed with 5% glucose until it reached to isotonic concentration, and the conductivity of the solution was determined by sterilized water in 5% glucose as the control, and it was marked as L1, and different volumes of MA solution were added into *S. aureus* suspension to reach the final concentrations of 31.25 µg/mL and 62.5 µg/mL. Then, the mixture was mixed and incubated at 37 °C, and the conductivity of the mixture was measured every 2 h and labeled as L2. The electrical conductivity of 5% glucose bacterial mixture was treated with boiling water for 5 min and marked as L0, and the relative electrical conductivity was calculated as = (L2−L1)/L0 × 100%. The suspension of *S. aureus* in the logarithmic growth phase was centrifuged for 10 min, and then the cells were resuspended and washed twice with PBS. Later, the concentration was adjusted to 10^7^ cfu/mL, and MA was added to reach the final concentration of 31.25 µg/mL and 62.5 µg/mL, but, for the control group, nothing was added. All the treated suspensions were incubated at 37 °C, and the bacterial solution was taken every 2 h for detecting the OD value in the medium supernatant at 260 nm and 280 nm and analyzing the content of AKP in the supernatant by AKP kit [37,38].

### 4.8. Effect of MA on Bacteria β- Determination of Galactosidase Content

For normal bacterial cells, *β*-galactosidase could not pass through the cell membrane, and, if the cell membrane was damaged, *β*-galactosidase could leak through the damaged membrane, and it could be detected in the cytoplasm [39]. The bacterial solution in the logarithmic growth period was centrifuged, resuspended by sterile M9 lactose induction medium, and incubated at 37 °C for another 8 h. After centrifugation, the cell pellets were resuspended by *β*-galactosidase reaction buffer, and 2-Nitrophenyl *β*-D-galactopyranoside (ONPG) was added and mixed thoroughly. Subsequently, MA was added to the suspension to reach the final concentrations of 31.25 µg/mL and 62.5 µg/mL, and nothing was added to the control group. All samples were incubated at 37 °C, and the cell suspensions were removed every 2 h for measuring OD values at 420 nm.

### 4.9. Effect of MA on Soluble Protein Content of S. aureus Detected by SDS-PAGE

The bacterial cells in the logarithmic phase were harvested by centrifugation at 5000× *g* for 10 min. and diluted by fresh sterilized LB medium to OD value of 0.5. Then, MA was added to the suspension for final concentrations of 31.25 µg/mL or 62.5 µg/mL, and the following incubation was set at 37 °C for 16 h. Subsequently, cells were harvested by centrifugation at 5000 r/min for 10 min and subjected to be disrupted by sonification on ice. Later, the cell lysates were centrifuged at 10,000× *g* and 4 °C for 10 min to collect the soluble fraction. The supernatants were finally subjected to 15% SDS-PAGE analysis, and the gels were stained with Coomassie brilliant blue R-250 and observed and imaged under the gel imaging system [34].

### 4.10. Determination of SDH and MDH Activities

Firstly, *S. aureus* cells that were intervened were 31.25 µg/mL or 62.5 µg/mL MA, marked as experimentational groups, and then they were intervened by DMSO, marked as a control group, which was then incubated, harvested, and disrupted by methods mentioned above. After sonification and incubation at 37 °C for 20 h, and centrifuging at 5000 r/min for 10 min, the activities of SDH and MDH in the supernatants were detected by the SDH and MDH activity detection kit [40].

### 4.11. Study on the Interaction Mode between MA and DNA

#### 4.11.1. Agarose Gel Electrophoresis Was Used to Determine the Interaction Mechanism between MA and DNA

A prokaryotic plasmid pET-28a DNA was intervened by added MA to the final concentrations of 0.003906, 0.007812, 0.15625l, 0.3125, 0.625, 1.25, and 2.5 mg/mL, respectively. For the control group, absolute alcohol was added to the system. All tubes were supplemented with buffer solution to reach 1.5 mL and subjected to constant temperature incubation at 37 °C for 30 min. Finally, the reaction system was terminated by adding 1 μL 10% SDS and 1 μL 10 mg/mL proteinase K. A further incubation was carried out at 37 °C for another 30 min, and then the samples were electrophoresed in 0.8% agarose gel electrophoresis for 20 min at 150 V, and then they were observed and photographed by a gel imaging system [41].

#### 4.11.2. The Interaction Mode between MA and DNA was Determined by UV Absorption Spectrometry

The reaction was carried out in the quartz cuvette. The pET-28a DNA was intervened by adding MA to the final concentrations of 31.25 µg/mL and 62.5 µg/mL and incubated at 37 °C for 30 min. Finally, the UV absorption spectrum was detected by a UV spectrophotometer at 240–300 nm. For the control group, absolute alcohol was added to the system [42].

### 4.12. Determination Method of the Influence of MA on DNA Topoisomerase Activity

The effect of MA on the activity of DNA topoisomerase was detected by gel electrophoresis [42]. The reaction system consisted of DNA unwinding buffer I/II,d superhelix pET-28a DNA, crude enzyme extract, and different concentrations of MA solution. DMSO was used to replace MA in the control group. After incubation at 37 °C for 30 min, the reaction was terminated by adding 10% SDS and 10 mg/mL proteinase K 1 μL, and a further incubation was carried out at 37 °C for 30 min. The samples were electrophoresed in 0.8% agarose gel electrophoresis for 20 min at 150 V. The gel was observed and photographed by gel imaging system.

### 4.13. Statistical Analysis

SPSS 22.0 statistical software was used to analyze the experimental data, and analysis of variance was used for pairwise comparison between groups. *p* < 0.05 was considered as significant correlation, and *p* < 0.01 was considered as extremely significant correlation.

## 5. Conclusions

MA plays a significant inhibitory role on *S. aureus* through destroying the integrity of cell membrane and cell wall integrity, affecting the changes of energy metabolism, inhibiting protein synthesis, interacting with DNA, and inhibiting the activities of DNA topoisomerase I and II. As a highly active antibacterial natural product with multiple targets to interfere with bacterial proliferation, metabolism, and cell integrity, it is very suitable for the development of low-resistance antibacterial drugs and can be used as a candidate molecule for the development of new antibacterial drugs. This study provides an important reference and basis for the follow-up treatment of *S. aureus* and other pathogenic bacteria infection and the development of low-resistance, multi-target antimicrobial drugs.

## Figures and Tables

**Figure 1 molecules-28-01895-f001:**
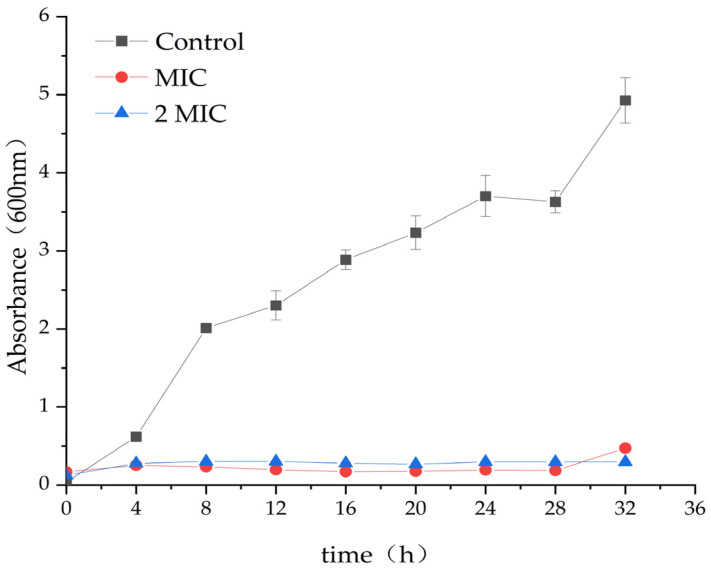
Effect of MA on the growth curve of *S. aureus*.

**Figure 2 molecules-28-01895-f002:**
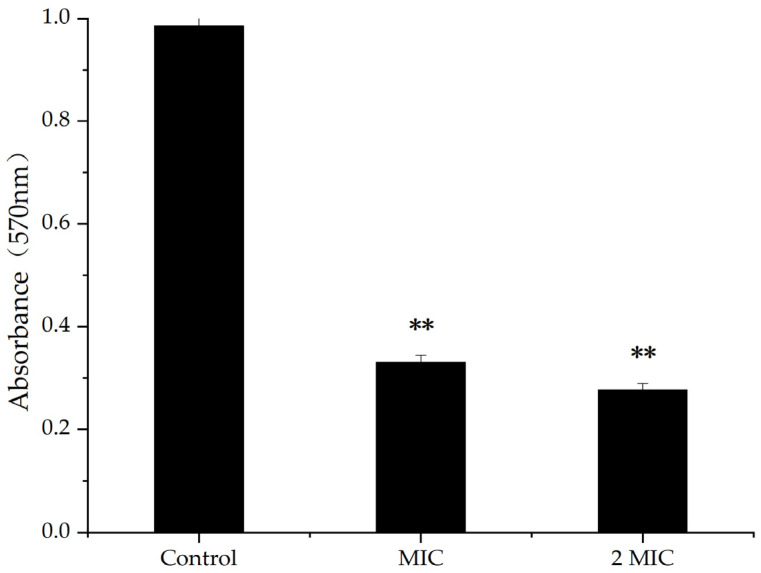
Inhibition of MA on biofilm; **: Compared with the control group, the difference is extremely significant (*p* < 0.01).

**Figure 3 molecules-28-01895-f003:**
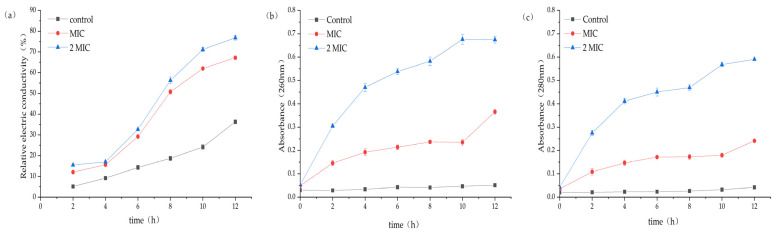
(**a**) Effect of MA on the electrical conductivity of *S. aureus* culture. (**b**) Effect of MA on Nucleic acid leakage of *S. aureus* culture. (**c**) Effect of MA on protein leakage of *S. aureus* culture.

**Figure 4 molecules-28-01895-f004:**
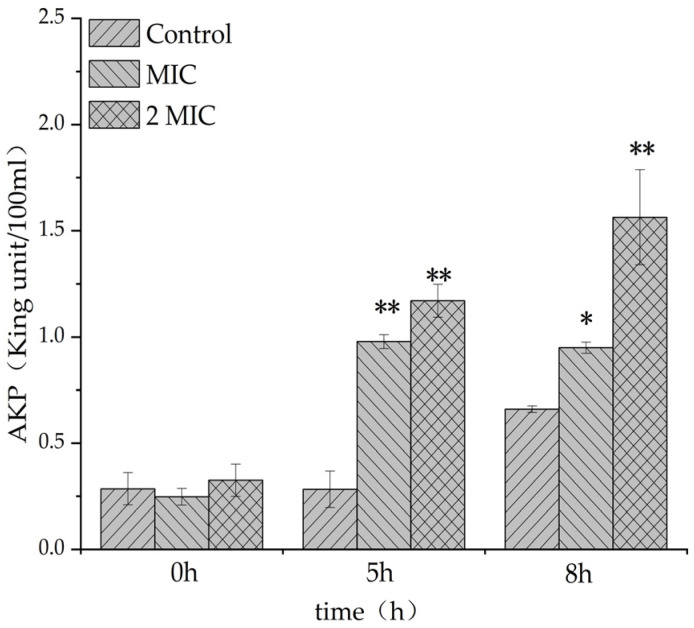
Effect of MA on the leakage of alkaline phosphatase from *S. aureus*. *: Compared with the control group, the difference is significant (*p* < 0.05); **: Compared with the control group, the difference is extremely significant (*p* < 0.01).

**Figure 5 molecules-28-01895-f005:**
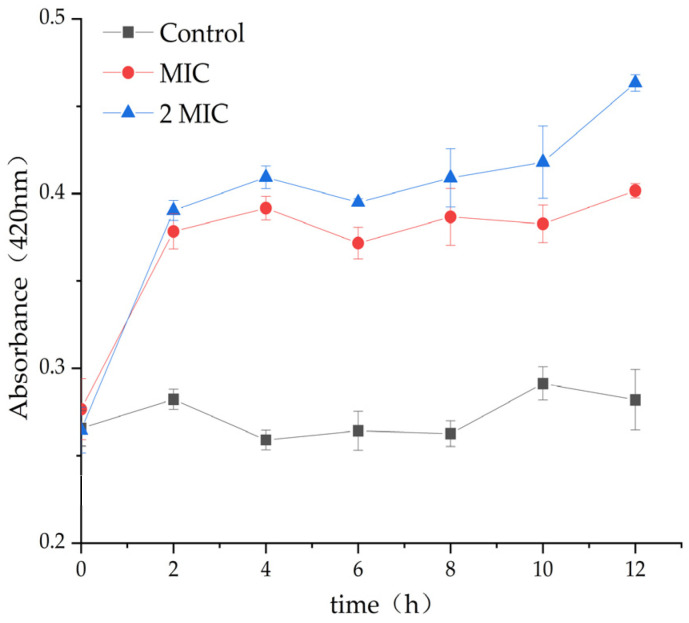
Effect of MA on the content of *S. aureus β*-galactosidase.

**Figure 6 molecules-28-01895-f006:**
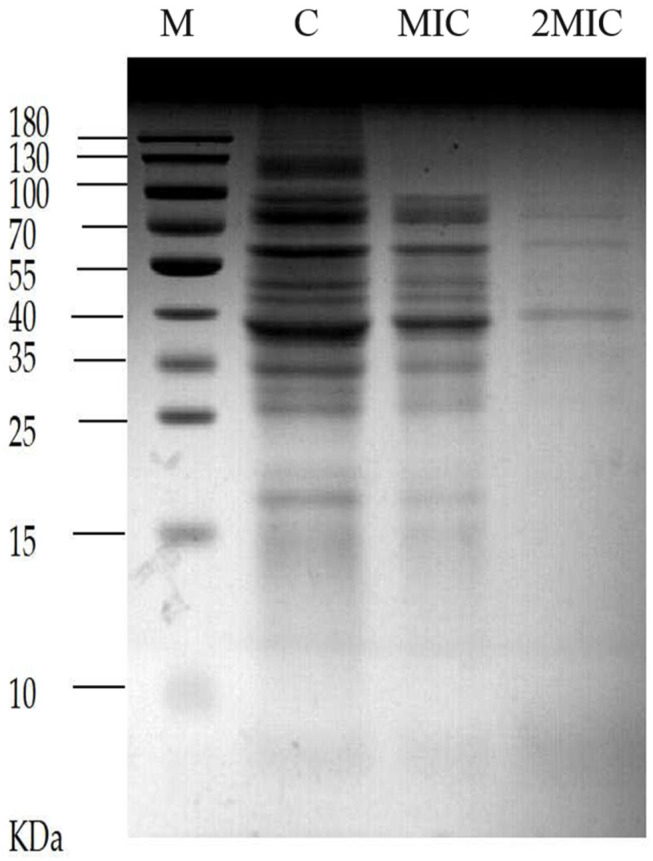
Effect of MA on soluble protein of *S. aureus*. Note: M = Marker, C = Control group.

**Figure 7 molecules-28-01895-f007:**
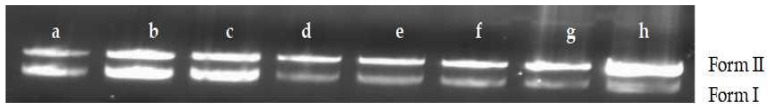
The direct break effect of MA on pET-28a DNA. Form II (open circular DNA and linear); Note: Form I (supercoiled DNA); a: 100% ethanol; b–h: 0.003906 mg/mL; 0.007812 mg/mL; 0.15625 mg/mL; 0.3125 mg/mL; 0.625 mg/mL; 1.25 mg/mL; 2.5 mg/mL.

**Figure 8 molecules-28-01895-f008:**
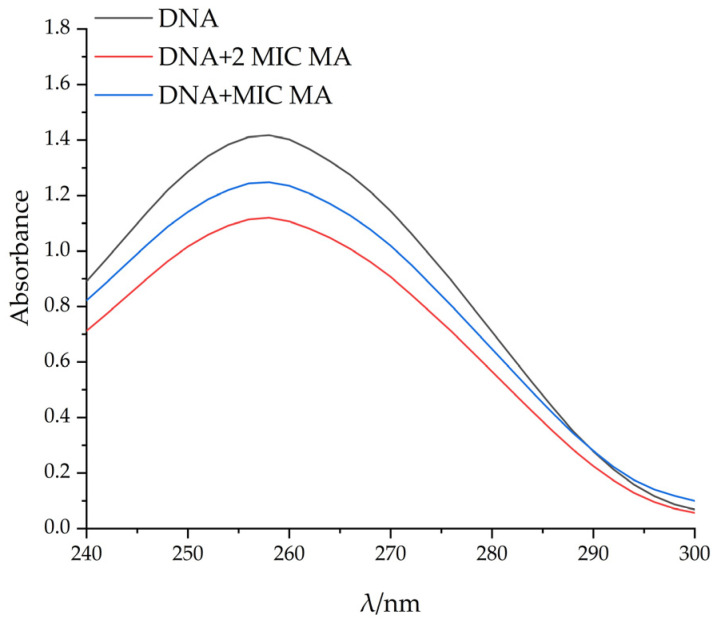
The absorption spectrometric titrations of MA with DNA.

**Figure 9 molecules-28-01895-f009:**
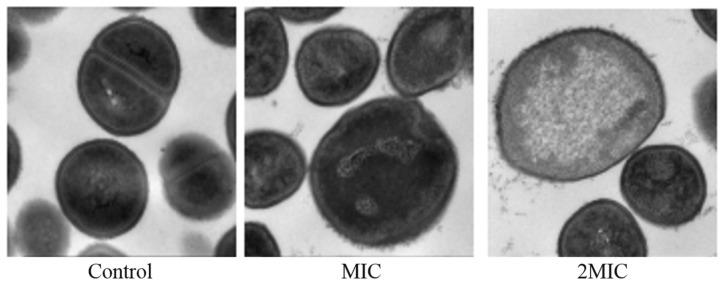
TEM images of *S. aureus* after treatment with MA.

**Figure 10 molecules-28-01895-f010:**
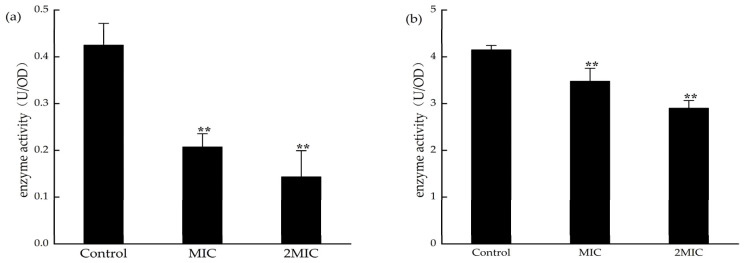
(**a**) Effect of MA on *S. aureus* SDH and (**b**) MDH; **: Compared with the control group, the difference is extremely significant (*p* < 0.01).

**Figure 11 molecules-28-01895-f011:**
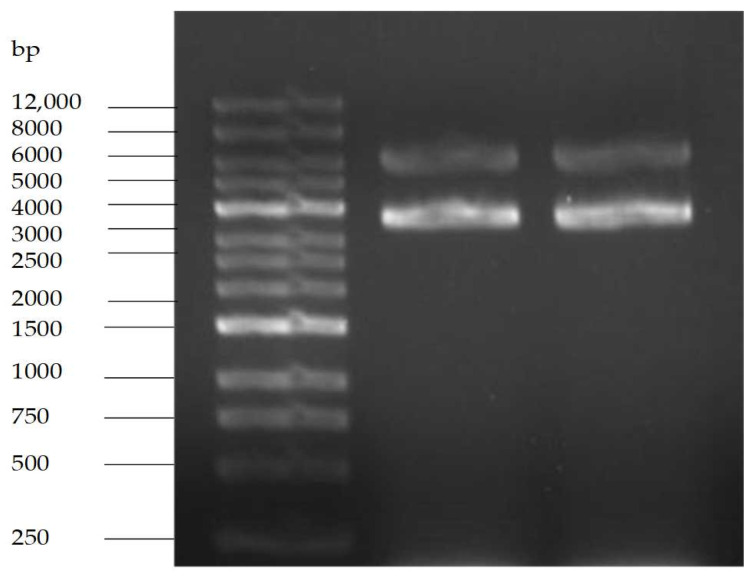
Agarose gel electrophoresis of pET28a.

**Figure 12 molecules-28-01895-f012:**
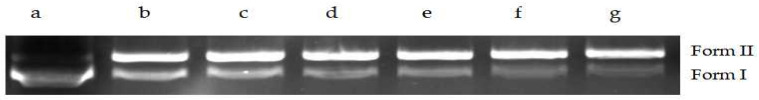
Effect of DNA topoisomerase on pET-28a DNA helicase. Note: Form II (open circular DNA and linear); Form I (supercoiled DNA); a: pET-28a; b–g: 2 µL; 2.8 µL; 3.6 µL; 4 µL; 6 µL; 8 µL DNA Topoisomerase extract.

**Figure 13 molecules-28-01895-f013:**
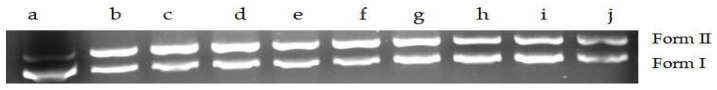
Effect of MA on topoisomerase I of *S. aureus*. Note: Form II (open circular DNA and linear); Form I (supercoiled DNA); a: pET-28a; b: topoisomerase; c: DMSO; d–j: 0.156 mg/mL; 0.3125 mg/mL; 0.625 mg/mL; 1.25 mg/mL; 2.5 mg/mL; 5 mg/mL; 10 mg/mL.

**Figure 14 molecules-28-01895-f014:**
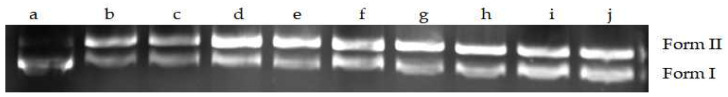
Effect of MA on topoisomerase II of *S. aureus*. Note: Form II (open circular DNA and linear); Form I (supercoiled DNA); a: pET-28a; b: topoisomerase; c: DMSO; d–j: 0.156 mg/mL; 0.3125 mg/mL; 0.625 mg/mL; 1.25 mg/mL; 2.5 mg/mL; 5 mg/mL; 10 mg/mL.

**Figure 15 molecules-28-01895-f015:**
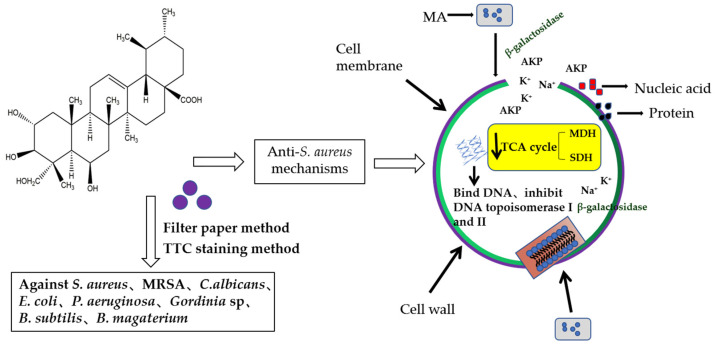
Antibacterial mode of MA aginst *S. aureus*.

**Figure 16 molecules-28-01895-f016:**
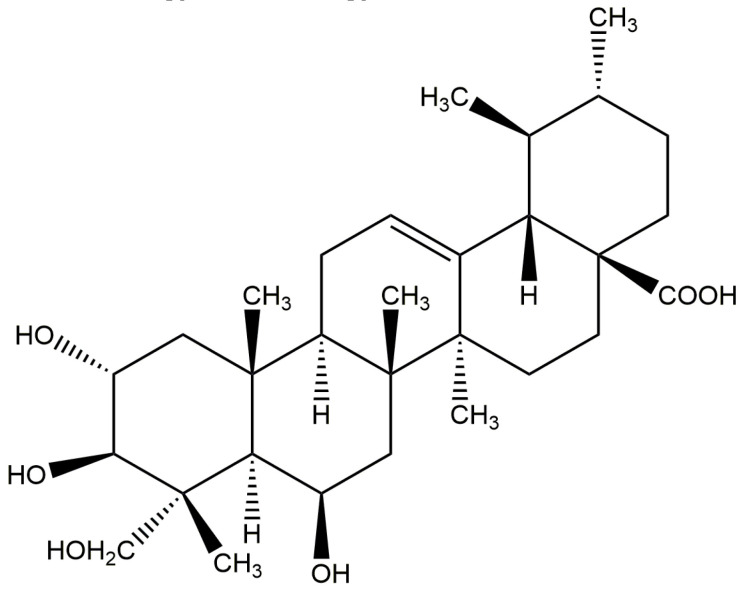
Chemical structure of MA.

**Table 1 molecules-28-01895-t001:** Diameter of inhibition zone (mm) and MIC (µg/mL) of MA against *E. coli*, *S. aureus*, MRSA, *Gordinia* sp., *P. aeruginosa*, *C. albicans*, *B. subtilis*, and *B. magaterium*.

Strains		Inhibition Zone (mm)	MIC (µg/mL)	
		MA	MA	OXA
Gram(−)	*E. coli*	10.0	250.0	/
Gram(+)	*S. aureus*	13.0	31.2	0.0
Gram(+)	MRSA	14.5	62.5	7.8
Gram(+)	*Gordinia* sp.	13.5	/	/
Gram(−)	*P. aeruginosa*	11.5	125.0	/
Fungus	*C.albicans*	14.0	/	/
Gram(+)	*B. subtilis*	10.5	62.5	3.9
Gram(+)	*B. magaterium*	10.5	62.5	1.9

Note: / means no experiment carried out.

## Data Availability

Personal information is included; thus, data are available for research only.

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
