# Peer review of "Antibacterial Activity and Mechanism of Madecassic Acid against Staphylococcus aureus"

_molecules, 2023, doi:10.3390/molecules28041895_

Round 1

Reviewer 1 Report

Minor remarks

·       All minor remarks are highlighted in the manuscript.

·       Authors should avoid the lumping of references in the manuscript, but each should be discussed.

Author Response

(1) Line 36: “infection and” should be “infection, and”.

Yes. We have revised this careless omission according to your comment.

(2) Line 86: we should provide the full name for abbreviation of “OD”.

Yes. We have added the full name for “OD” here and the line No. has been changed to line 100.

(3) Line 93: we should add “A” in at the beginning of this paragraph.

Yes. We have added “A” here and the line No. has been changed to line 108.

(4) Line108: the word “suggest” should be “suggests”.

Yes. We have revised this careless omission according to your comment here and the line No. has been changed to line 124.

(5) Line142: the phrase “From the results showed in ” should be “The results shown in”.

Thanks for your advice, but we think the sentence here should be revised to “From the results shown in Figure 6, which were analyzed…….”. The line No. has been changed to line 159.

(6) Line187: the word “suggest” should be “suggests”.

Yes. We have revised this careless omission according to your comment here and the line No. has been changed to line 210.

(7) Line 227: “breaking and” should be “breaking, and”.

Yes. We have revised this careless omission according to your comment here and the line No. has been changed to line 253.

(8) Line 285: “of control” should be “of the control”.

Yes. We have revised this careless omission according to your comment here and the line No. has been changed to line 318.

(9) Line 371: “concentration ethanol” should be “ethanol concentration”.

Yes. We have revised here according to your comment and the line No. has been changed to line 421.

(10) Line 400 and line 407: “Subsequently” should be “Subsequently, ”.

Yes. We have revised this careless omission according to your comment and the lines No. have been changed to line 451 and line 460, respectively.

(11) Line 402: “The all the samples” should be “All samples”.

Yes. We have revised here according to your comment here and the line No. has been changed to line 453.

(12) Line 435: “Reaction” should be “The reaction”.

Yes. We have revised this careless omission according to your comment here and the line No. has been changed to line 486.

(13) Line 488: we should delete the needless “[7]”.

Yes. We have revised this careless omission according to your comment here and the line No. has been changed to line 549.

Reviewer 2 Report

The manuscript “Antibacterial Activity and Mechanism of Madecassic Acid 2 against Staphylococcus aureus” is, overall, well written, but there are some shortcomings on which the authors should respond:

1)      P 1: Line 43 and 44 assert that to replace antibiotics, especially to mine potential natural
compounds with significant anti-microorganisms activity from pharmaceutical botany.
”… It is a bit premature to talk about the “replacement of antibiotics” in human therapy…… many studies report the possibility of using natural products in synergistic combinations with antibiotics as adjuvants in antibiotic therapy, in particular against antibiotic-resistant bacteria. The association of natural compounds with antibiotics, may be a possible strategy to fight resistant strains: phytoconstituents can perform as bacterial resistance-modifying agents, restoring the effectiveness of commercial drugs. This may be due to the different mechanisms of action of natural products.

The sentence needs to be rephrased.

2)      P 3: Lines 87-89 assert that 31.25 μg/mL MA could completely inhibit the growth of S. aureus within 28 h. The growth of S. aureus was completely inhibited within 32 h after treatment with 62.5 μg/mL MA (Figure 1).MIC is faster (28h) than 2 MIC (32h) to completely inhibit the growth of S.a.? Moreover, Figure 1 shows a re-growth of S.aureus in the red line at 32h  and 31.25 μg/mL MA. For this concentration erase “completely” and reword the sentence.

3)      A lot of scientific names of bacteria are not italicized: P.  Line 103; P  8 lines 201 and 211; P 9 line 224; P 11 lines 314-317.

4)      P 9 line 246: ”Zhou et al [25]. found that “…..Zhou et al. [25] found that…..

5)      The resolution of figures 2 and 3 needs to be improved

Author Response

(1) P 1: Line 43 and 44 assert that "to replace antibiotics, especially to mine potential natural compounds with significant anti-microorganisms activity from pharmaceutical botany."... It is a bit premature to talk about the "replacement of antibiotics" in human therapy...... many studies report the possibility of using natural products in synergistic combinations with antibiotics as adjuvants in antibiotic therapy, in particular against antibiotic-resistant bacteria. The association of natural compounds with antibiotics, may be a possible strategy to fight resistant strains: phytoconstituents can perform as bacterial resistance-modifying agents, restoring the effectiveness of commercial drugs. This may be due to the different mechanisms of action of natural products

The sentence needs to be rephrased

Thank you very much for your kindly guidance. We have rephrased this sentence in line 42 to line 49.

(2) P 3: Lines 87-89 assert that “31.25 μg/mL MA could completely inhibit the growth of S. aureus within 28 h. The growth of S. aureus was completely inhibited within 32 h after treatment with 62.5 μg/mL MA (Figure 1)." MIC is faster (28h) than 2 MIC (32h) to completely inhibit the growth of S.a.? Moreover, Figure 1 shows a re-growth of S.aureus in the red line at 32h and 31.25ug/mL MA. For this concentration erase “completely” and reword the sentence.

Yes, the expression here should be modified according to your meticulous advices. We have revised it and the line No. has been changed to line 99 to line 104.

(3) A lot of scientific names of bacteria are not italicized: P. Line 103; P 8 lines 201 and 211; P 9line 224; P 11 lines 314-317.

Yes. We have revised this careless omission according to your comment and checked the whole manuscript. The lines No. have been changed to line 119, line 126, line 136, line 146, line 172, line 184, line 205, line 216, line 224, line 235, line 250, line 342 to line 354.

(4) P 9 line 246: “Zhou et al [25]. found that” .....Zhou et al. [25] found that....

Yes. We have revised this careless omission according to your comment here and the line No. has been changed to line 276.

(5) The resolution of figures 2 and 3 needs to be improved.

Yes. We have replaced these two figures with the high resolution original figures.

Reviewer 3 Report

Please check the attached file!

Author Response

(1) The manuscript of Wei et al. decribes investigations into antibacterial activity of the pure compound madecassic acid (MA), as an active principle of the TCM drug Centella asiatica. Using different biochemical and molecular biological methods an approach to mechanistic aspects of the antibacterial effect of MA was made. Since natural products are interesting leads for the search of new antibiotics, to overcome resistance, the manuscript is interesting for other researchers.

However, I miss comparison of these investigations with literature reports on antibacterial activities of other (isolated) triterpenoids (e.g. of the ursane or oleane type, et.c.). The authors may include one or two review articles on this topic?

Thank you very much for your thoughtful advice. We have added the antibacterial activity comparison between MA and the other two reported triterpenoid compounds (tormentic acid and sanguisorbigenin). The revised content was in the line 89-94.

(2) Moreover, current literature on bacterial multidrug efflux pumps (or MDR pump inhibition) are lacking, since research on targeting bacterial MDR pumps by natural products is a topic issue. From my point of view it could be included in the discussion or introduction (for an example see Stermitz et al. 2000, Proc. Natl. Acad. Sci. U S A, 97(4):1433-7 or doi: 10.1073/pnas.030540597).

Yes. We have added the corresponding discussion in line 342-353.

(3) Specific comments:

Please have grammar and orthography checked by a native speaker.

Page 1, line 19: MA had an inhibitory effect …

Page 1, line 22: …, respectively. For instance, 31.25 µg/mL MA …

Page 1, lines 42-44: The phrase anti-microorganism activity “is a little unfortunate. Please replace or specify!

Yes. We have revised this careless omission according to your comment and marked in red color fonts in the relative place in our manuscript.

(4) Page 2, lines 45-47: Is the distribution of C. asiatica limited to China, or can it be found in other countries (where also?).

Yes, it can also be found in other countries, we have added the corresponding content in line 52-54.

(5) Page 2, line 48: What do you mean with whole grass? C. asiatica is a dicotyledonous plant and not a grass. Please check!

Sorry, it is a Chinese style English translation. It means the whole plant. We have revised it in line 54.

(6) Page 2, line 67: ...in the dimethyl sulfoxide (DMSO, control) group.

Yes. We have revised this careless omission according to your comment here and the line No. has been changed to line74.

(7) Page 2, Table 1: Could you specify for the reader in the Table which bacteria are Gram(+) or (-), e.g. by footnote?

Yes. We have added the corresponding content in Table 1.  

(8) Page 2, Table 1, column 3: Please reduce/specify all values on one decimal place (250.0, 31.2, 125.0 …e.c.).

Yes. We have revised this careless omission according to your comment.

(9) Page 3, line 84: The growth of S. aureus in the control group was good … (This is an evaluation!?) … Better: In comparison to the control group (DMSO) the treatment with 31.25 µg/mL and … MA significantly inhibit the growth of S. aureus.

We are so sorry for inaccurate expression here and have revised the expression according to your comment in line 99-100. Thank you for your kindly help.

(10) Page 3, chapter 2.3.: I am a little sceptical on this biofilm assay. If you cause an antibacterial (growth) effect by treatment with doses of MA, than logically you also disturb the “biofilm” formation on the glass surface? Please discuss it! The term “biofilm” in this context is a little exaggerated since it is more often used to describe a symbiotic community of different bacteria species?

Thanks for your question. Biofilm is consisted of bacterial extracellular macromolecules, which refers to the organized bacterial population which is attached to the surface of living or inanimate objects and can be secreted by both one type and multi-type microorganisms for cell colonization or protecting the bacterial cells. The spectrophotometry method is more sensitive than microscope analysis on the glass surface, which can also calculate an accurate value for the analyzed sample.

  • Page 5, chapter 2.5, Figure 6: Please specify in the legend what are M, C, MIC and 2MIC. I also miss a list of abbreviations. Why did you apply SDS-Page and not the more common Bredfort or BSA method to determine soluble protein?

Thanks for your question and advice. We have added the full name for “M”, “C” in Fig. 6. And we also added the list of abbreviations in our manuscript in line 517-524.

For the second question, the more common Bredford or BSA method is usually used to determine the purified protein samples or samples with less disturbing components; for evaluating the expressed protein, SDS-PAGE is a more common method than Bredford or BSA method, which can give a visual protein bands image for quickly evaluating the express level of targeted proteins.

(12) Page 6, Figure 7: Form II and Form I please line up with the corresponding DNA bands.

Yes. We have revised this careless omission according to your comment here.

(13) Page 7, chapter 2.8, Headline: What are MDH and SDH? Abbreviations list?

Yes. We have added the full name for “MDH and SDH” in line 205-206.

(14) Pages 8-9, Chapters 2.9.2 – 2.9.4.: A concentration dependent effect of MA on pET28a DNA helicase is obviously recognizable from the Western blot, but no effect on topoisomerase I and II of S. aureus. Could you please check if you properly discussed these results?

Thanks for your question. In Fig.13, along with the concentration increasing of MA, the brightness of Form I pET28a are also increasing, and the brightness of Form II pET28a turn less brightness gradually, indicating that MA has an inhibition activity on Staphylococcus aureus topoisomerase I. Meanwhile, the similar phenomenon can be seen in Fig 14, indicating that MA also has an inhibition activity on Staphylococcus aureus topoisomerase II. Perhaps your question focuses on the “a”, “b”, “c” lanes in Fig 14 showing darkness bands, which is caused by the loading amount of plasmids samples. Wish you can understand our explanation.

(15) Page 11: line 332: What is “vernier caliper”, an imaging system? Please explain for uninitiated readers.

Vernier caliper is an accurate ruler for measuring length, inner and outer diameter and depth. We have added the corresponding explanation in the line 381-382.